# Women's multidimensional empowerment index and essential newborn care practice in Bangladesh: The mediating role of skilled antenatal care follow-ups

**Kanchan Kumar Sen***, **Ahsan Rahman Jamee**, **Wasimul Bari**

Department of Statistics, University of Dhaka, Dhaka, Bangladesh

* kksen.sta@du.ac.bd

## Abstract

### Introduction

The World Health Organization recommended a set of five neonatal care practices that are most essential for good health of a newborn. As good essential newborn care (ENC) practice reduces the risk of neonatal morbidity and mortality, this paper focuses how women's multidimensional empowerment index is associated with this practice through the skilled antenatal care. To the best of knowledge, no such study has been conducted yet. The composite index of women's multidimensional empowerments was constructed using family decision, intimate partner violence, social status, healthcare access and economic status of women; and skilled antenatal care was defined if a pregnant woman received eight or more checkups during pregnancy from skilled health professionals.

### Materials and methods

Data extracted from Bangladesh Demographic and Health Survey, 2017–18 have been utilized in the study. A total of 2441 mothers have been considered who delivered their last live birth at home within three years preceding the survey. A mediation analysis was performed considering the structural equation modeling to find out the adjusted association of women's empowerment on both skilled antenatal care and good ENC practice, but the unadjusted associations were also checked using a chi-square test. To test the indirect as well as total effect of women's empowerment through skilled antenatal care on good ENC practice, bias-corrected standard errors were estimated using a bootstrapping sampling.

### Results

Good ENC practice was considerably low in Bangladesh with 7.6% of newborns receiving the practice. Both adjusted and unadjusted analyses showed the significant association of women's empowerment with both skilled antenatal care and good ENC practice. The study revealed that the indirect effect of high empowerment through skilled antenatal care was greater than its direct effect on good ENC practice.

**Data Availability Statement:** Data are available in a public, open access repository. Data are available on the website (https://dhsprogram.com/data/

dataset/Bangladesh_Standard-DHS_2017.cfm?
flag=0).

**Funding:** The author(s) received no specific
funding for this work.

**Competing interests:** The authors have declared
that no competing interests exist.

## Conclusion

This study explored that the prevalence of good ENC practice can be accelerated through
women's empowerment, where skilled antenatal care plays an important mediating role in
improving good ENC practice among highly empowered mothers. The study suggests that a
woman should follow the latest guidelines recommended by WHO for antenatal care follow-
up. Policymakers can modify some of the maternal and child health care interventions
based on the research findings.

## Introduction

Neonatal mortality shares 47% of the global under-5 deaths, where one-third of these deaths
were observed on the day of birth [1]. Moreover, the first week of life is the most vulnerable
period [2] as nearly 75% of neonatal deaths occurred during that time [1,3]. A substantial
improvement has been made worldwide regarding child survival since 1990. The global rate of
neonatal deaths was decreased by 52% from 1990 to 2019; however, this decline has been
slower compared to the under-five child mortality rate [1]. In Bangladesh, neonatal deaths
contributed around 67% of all under-five mortality [4]. To achieve the Sustainable Develop-
ment Goal (SDG) 3.2, which aims to reduce the neonatal mortality rate to 12 per thousand live
births by 2030, Bangladesh lags far behind this target as the current neonatal mortality rate is
30 per 1000 live births [4,5]. The majority of neonatal deaths occur due to premature birth,
intrapartum conditions of labor, infections, or imperfect manner of newborn care practices
[6–9]. Essential newborn care (ENC) practice and supervision of skilled health personnel dur-
ing labor and childbirth may prevent about 70% of neonatal deaths [10,11]. Therefore, to
improve the health status of newborns and to reduce the neonatal mortality to a great extent,
appropriate strategies need to be developed for the essential newborn care practice.

Regardless of place of delivery or size of the infant, every newborn baby needs essential
newborn care (ENC). World Health Organization (WHO) recommends five practices for
ENC focused on using clean instruments to cut the umbilical cord, applying nothing to the
cord, immediate drying (within 5 minutes), delayed bathing (72 hours after birth), and imme-
diate initiation of breastfeeding (within 1 hour of delivery) [4,12]. A large number of neonatal
deaths occur due to bacterial infections to the umbilical cord. However, these infections can be
prevented and thus reduce the risk of neonatal mortality by practicing proper care to the
umbilical cord [13,14]. Hypothermia increases neonates' intracranial hemorrhage and sepsis
which further affects infant mortality [15]. To reduce the risk of hypothermia, newborns
should be dried and put on the mother's bare chest within minutes after birth, and they should
be bathed late [16–18]. Initiating immediate breastfeeding also alleviates the risk of diarrhea,
pneumonia, neonatal sepsis, and meningitis [19].

Among non-institutional births, the risk of neonatal mortality is higher compared to the
facility births; and the prevalence of practicing essential newborn care is very low [6,20]. Also,
in developing countries like Bangladesh, mothers of home-born children had inadequate
knowledge regarding the practice of essential newborn care. The majority of live births were
non-institutional and delivered in the absence of skilled personnel; and such mothers had
insufficient access to proper antenatal care [21]. The WHO recommended at least eight ante-
natal checkups by the skilled attendant to reduce perinatal mortality and improve the experi-
ence of newborn care for pregnant women [22,23]. Skilled antenatal care (SANC) providers

also guide mothers to acquire knowledge about essential newborn care practices and maintaining healthy conditions during pregnancy [24]. Moreover, women having non-institutional birth are likely to seek services from traditional birth attendants rather than skilled providers, which may result in high maternal and neonatal deaths [25].

Women's empowerment can inspire pregnant women to utilize skilled ANC services [25–27]. Empowered women have also more access to the modern health facilities, which in turn helps to reduce neonatal mortality [28–31]. Recent studies revealed that a low rate of women's empowerment and limited access to receive skilled ANC increase the risk of neonatal mortality [32–34]. However, most of the studies employed only one or two dimensions to describe women's empowerment, concentrating on family decision-making autonomy and attitudes toward intimate partner violence [35] that limit women's empowerment. Another study proposed a survey-based women's empowerment index (SWPER) for African countries, which included three dimensions: social independence, decision making, and attitude toward violence [36]. A recent study assessed women's empowerment across five domains: health care access, socio-economic status, decision making autonomy, attitude toward violence, and asset ownership [35]. As a result, the current study, like the previous two, used a multidimensional women's empowerment index to emphasize empowerment from multiple perspectives in Bangladesh [35,36].An attempt has been made in this study to examine how empowerment of women plays a role in improving the ENC practices directly as well as indirectly through the utilization of skilled antenatal care. To best of our knowledge, this was the first study that developed a composite index for women's empowerment by assessing the five dimensions such as women's autonomy in family decisions, attitudes toward wife-beating by husband, barriers faced by women in accessing health care, women's social independence and women's assets using a weighting approach followed by Alkire and Foster [37]. Latest guideline of the WHO on receiving antenatal care checkups from skilled health personnel has been followed to define SANC. Mediation technique for examining the mediating role of skilled antenatal care to strengthen the relationship between women's empowerment and ENC practice has also been adopted [38]. For the purpose of the analysis, data have been extracted from Bangladesh Demographic and Health Survey (BDHS) 2017–18.

## Conceptual framework

The women's empowerment is an important indicator for achieving some sustainable development goals because it contributes in improving family health outcomes, including child health [35,39]. Empowered women are more likely to have better access to healthcare and have control over health resources that improve their health as well as the health of their families and children [35,40–45]. Previous studies also examined that women's empowerment play a vital role in utilizing the maternal healthcare services such as antenatal care, delivery care and postnatal care [46–48]. Again, antenatal care visits are significantly associated with essential newborn care practices [49–52]. In this study, it has been hypothesized that women's multidimensional empowerment index (MEI) positively influences the ENC practice, as well as skilled ANC follow-ups; and skilled ANC also play an important role in enhancing the ENC practice. The conceptual path diagram is shown in Fig 1.

## Materials and methods

### Data source

The study used a recent and nationally representative cross-sectional data obtained from the Bangladesh Demographic and Health Survey (BDHS), 2017–18 to address the objectives of this study. The National Institute of Population Research and Training (NIPORT) collected

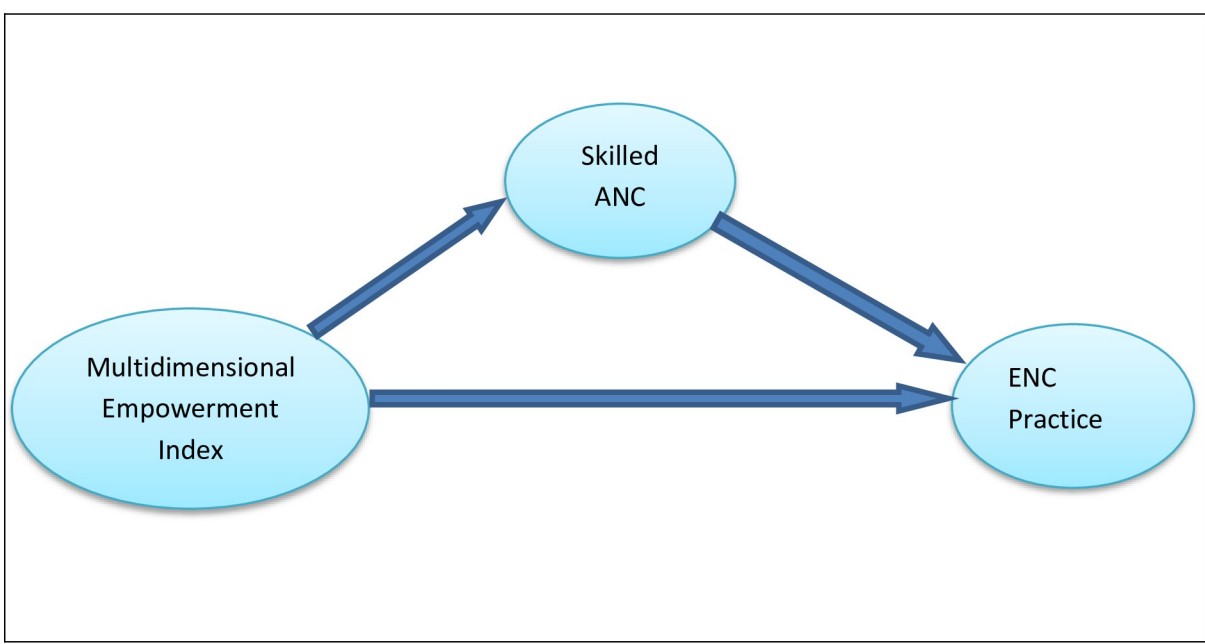

**Fig 1. Conceptual path diagram.**

data from October 2017 to March 2018 in collaboration with the DHS program and the United States Agency for International Development (USAID)/Bangladesh. The survey provided information on various health and demographic indicators such as maternal and child health, fertility, family planning and nutrition etc.

## Study design

The sampling frame of the 2011 Population and Housing Census of the People's Republic of Bangladesh provided by the Bangladesh Bureau of Statistics (BBS) was used to collect the data in BDHS, 2017–18. The survey was conducted following two-stage stratified sampling, where a total of 675 enumeration areas (EA), known as clusters, were selected in the first stage with probability proportional to size (PPS) and an average of 30 households were randomly chosen from each selected EA in the second stage using a systematic random sampling. Due to the massive flood in three EAs, a total of 672 clusters were successfully surveyed which included a total of 20,127 ever-married reproductive women. Details of the sample design can be found in the BDHS, 2017–18 final report [4]. Furthermore, as the BDHS, 2017–18 data were collected using a complex sample structure, adjustments for the complex sampling structure were taken care of using the survey Stata command (*svy*).

## Participants

The primary objective of this study was to explore the ENC practices among the home births as the BDHS, 2017–18 only recorded the essential newborn care practices for home births. In the original data set, the information on ENC indicators for institutional births were not available. However, the study selected the reproductive married women who gave the last birth at home within three years preceding the survey. Finally, a total of 2479 women were selected after excluding missing values in this study.

### Ethics approval and consent to participate

The 2017–18 Bangladesh Demographic and Health Survey was implemented under the authority of National Institute of Population Research and Training (NIPORT). The data collection procedures were approved by the Institutional Review Boards of the ICF International, Rockville, MD, USA and Bangladesh Medical Research Council, Dhaka, Bangladesh. All respondents signed consent forms to participate in the study BDHS, 2017–18. Respondents who did not provide consent were not included in the analysis for the present study. All methods were carried out in accordance with relevant guidelines and regulations.

### Outcome measure

Essential newborn care (ENC) practice includes a set of initial cares for babies after birth: cord care (use of clean delivery kit/bag or boiled blade to cut the umbilical cord), nothing applied to the cord or only chlorhexidine applied, immediate drying the baby within 5 minutes, delaying bathing until 72 hours of birth and immediate breastfeeding within 1 hour of birth [4]. According to WHO and National Neonatal Health Strategy and Guidelines for Bangladesh, these practices are recommended to save a newborn [4,16,53]. The outcome variable, ENC practice, was considered as binary random variable, where 1 indicates good ENC and 0 for the poor ENC. A newborn is said to be received good ENC if all five components were practiced; whereas the practice is poor if he/she missed any of the components.

### Main exposure: Multidimensional Empowerment Index (MEI)

The MEI is a composite index consisting of several dimensions related to women's empowerment, with each dimension having multiple indicators of empowerment. In the study, we considered five dimensions to measure the index: women's autonomy in family decisions, attitudes toward wife-beating by husband, barriers faced by women in accessing health care, women's social independence and women's assets. A significant measure of women's empowerment is asset ownership. The information on "owns land alone or jointly with husband" and "title deed on land owned by women" that was utilized as asset ownership in another study to determine empowerment was absent in the BDHS, 2017–18 data [35]. Due to the lack of availability of this information in BDHS, 2017–18, household assets were regarded as women's assets in the study. At present, women contribute to their households in improving their assets in Bangladesh. Only women are allowed to join a variety of NGOs, including Grameen Bank, ASA, Brac, Proshika, and others, and they are also only permitted to apply for loans to fund their families or to run small businesses. The BDHS 2017–18 study states that nearly half of women who have ever been married are actively employed, and their contributions to the growth of household assets are also significant. To build their families, each woman in a household has physical as well mental contributions. However, previous studies used a principal component analysis (PCA) [36,46] or a factor analysis [35,54] to measure the women's empowerment index, where the studies developed several factors (or PCs) for MEI, but we have used a weighting approach giving equal weights to each dimension for getting a composite index of women's empowerments following a technique of multidimensional poverty measures developed by Alkire and Foster and other research in the current study [37,55]. This composite index assigns a score to each woman, with higher scores signifying greater empowerment. The score ranges from 0 to 1, with 0 signifying little empowerment and 1 representing complete empowerment. We can easily discern the amount of women's empowerment based on the composite weighting score. Details about the dimensions, indicators, cut-offs, and weights of women's MEI have been shown in Table 1. Since we have considered five domains,

**Table 1. Dimensions, indicators, cut-offs and weights of women's multidimensional empowerment index (MEI), BDHS, 2017–18.**

| Dimension | Indicator | Empowered if woman (cut-off) | Weight (indicator) | Weight (Dimension) |
|---|---|---|---|---|
| Household decision-making autonomy | Own healthcare | takes decision alone or jointly with her husband. | 0.050 | 0.20 |
| | Household purchases | takes decision alone or jointly with her husband. | 0.050 | |
| | Family visits | takes decision alone or jointly with her husband. | 0.050 | |
| | Husband's earnings | takes decision alone or jointly with her husband. | 0.050 | |
| Attitude towards intimate partner violence | Wife beating justified: goes out without telling the husband | does not agree. | 0.040 | 0.20 |
| | Wife beating justified: neglects the children | does not agree. | 0.040 | |
| | Wife beating justified: argues with husband | does not agree. | 0.040 | |
| | Wife beating justified: refuses to have sex with husband | does not agree. | 0.040 | |
| | Wife beating justified: burns the food | does not agree. | 0.040 | |
| Health access barriers | Permission to go to the doctor for seeking medical advice/treatment | gets permission | 0.050 | 0.20 |
| | Money for treatment | gets | 0.050 | |
| | Distance to health facility | does not think as a problem | 0.050 | |
| | Not wanting to go alone to seek medical advice/ treatment | does not think as a problem | 0.050 | |
| Social independence | Woman's Education | completes secondary or higher education. | 0.025 | 0.20 |
| | Frequency of reading newspaper/ magazine in a week | reads once or more. | 0.025 | |
| | Frequency of watching television in a week | watches once or more. | 0.025 | |
| | Has an account in a bank | has. | 0.025 | |
| | Employment | has earnings from her works | 0.025 | |
| | Age of woman at first birth | is of age 19 or more years | 0.025 | |
| | Age at marriage | is of age 18 or more years | 0.025 | |
| | Spousal education difference | has equal or higher education than her husband. | 0.025 | |
| Asset ownership | Household assets | possesses two or more of the following stuff: water pump, air conditioner, computer, mobile telephone, radio, TV, refrigerator, bike or owns a car/truck. | 0.200 | 0.20 |
| Total | | | 1 | 1 |

one-fifth (0.2) of the total weight has been given to each domain and the weights of the dimensions have also been equally divided among the indicators of respective dimension.

A weighted arithmetic mean was calculated to get the individual score of empowerments. For computing the score, all the indicator variables were converted to binary variables indicating 1 for a woman who was empowered and 0 otherwise on the indicators of empowerment. Mathematically, the individual score can be viewed as

$$Z_i = \left( \frac{1}{m} \sum_{j=1}^{m} \frac{1}{p_j} \sum_{k=1}^{p_j} y_{ijk} \right);$$

where $Z_i$ is the score of the $i^{th}$ ($i = 1,2,\ldots,n$) individual (woman), $m$ is the number of dimensions, $p_j$ is the number of indicators in the $j^{th}$ dimension, $y_{ijk}$ is the value of the $k^{th}$ indicator in the $j^{th}$ dimension for the $i^{th}$ woman, and $n$ is the number of total individuals [56]. Theoretically, the score of a woman lies between 0 and 1, where 0 indicates that the woman is not

empowered and 1 indicates fully empowered at each dimension. Therefore, it can be said that as the score increases, so does the empowerment of woman. To examine the effect of MEI on ENC practice at different levels of scores, MEI was divided into three groups: low for scores below 0.50, average for scores between 0.50 and 0.75, and high for scores of 0.75 or higher.

## Mediator: Skilled Antenatal Care (SANC)

Pregnant women need antenatal care (ANC) check-ups from skilled health personnel to reduce pregnancy complications. Again, an optimum number of ANC visits provide good health for both mother and newborn. Skilled health personnel are called those health professionals who are either doctors, nurses, midwives, family welfare visitors, or community skilled birth attendants [4]. In 2016, the WHO recommended eight or more ANC visits for a pregnant woman [22,57]. However, the skilled antenatal care (SANC) was defined as the antenatal care received from skilled health professional in the study. By taking the recent WHO recommendation on ANC follow-ups into account, the variable, SANC, was categorized into two groups: 8+ SANC visits and <8 SANC visits, and this variable was used as a mediator to explain the relationship between MEI and ENC practice in the study. Note that the recent WHO guideline on ANC visits (2016) was followed in this study to observe how eight or more ANC visits influence good essential newborn care practice. Despite the fact that many of the births and related ANC visits (or non-visits) included in the study occurred prior to the revised guidelines, this study will provide more evidence to follow the most recent WHO guideline on ANC visits to ensure appropriate ENC practice. Furthermore, the current WHO recommendation on ANC visits was also used in the literature, which included data from the 2018 Demographic and Health Survey [58–60].

## Control variables

Following the previous studies, women's current age, birth order, place of residence and region were considered as control variables in the study [8,21,61]. The current age was categorized into three categories: 15–20, 20–30 and 30+ years; birth order: first, second/third and fourth/ higher; gender of index child: male and female; place of residence: rural and urban. The region was constructed from the eight administrative divisions of Bangladesh, and we categorized the region into three categories: eastern (Chittagong and Sylhet), central (Barisal, Dhaka and Mymensingh) and western (Khulna, Rajshahi and Rangpur).

## Statistical analysis

The data extracted from BDHS, 2017–18 were prepared for the analysis after excluding few missing values, and all analyses were conducted using Stata 17. The descriptive statistics of the selected variables were given as frequency percentages. To examine how the mediator and the selected covariates are associated with the ENC practice, analysis of measure of unadjusted association was conducted with chi-square test.

As one of the major objectives in the study was to explore the indirect effect (i.e., mediating effect) of MEI on ENC practice through the skilled ANC visits, a mediation analysis was performed to examine the hypothesis [38]. Hence, a structural equation modeling technique was used by setting two equations for getting the direct effects. The mediator variable, SANC, was modeled in the first equation considering MEI and other variables as covariates; and the outcome variable, ENC practice, was also modeled in the second equation considering MEI, SANC and all other variables as covariates. As the observations on SANC or good ENC practice are likely to be correlated within each cluster, a multilevel binary logistic regression model was employed to address the cluster effect for both equations. To check the cluster effect on

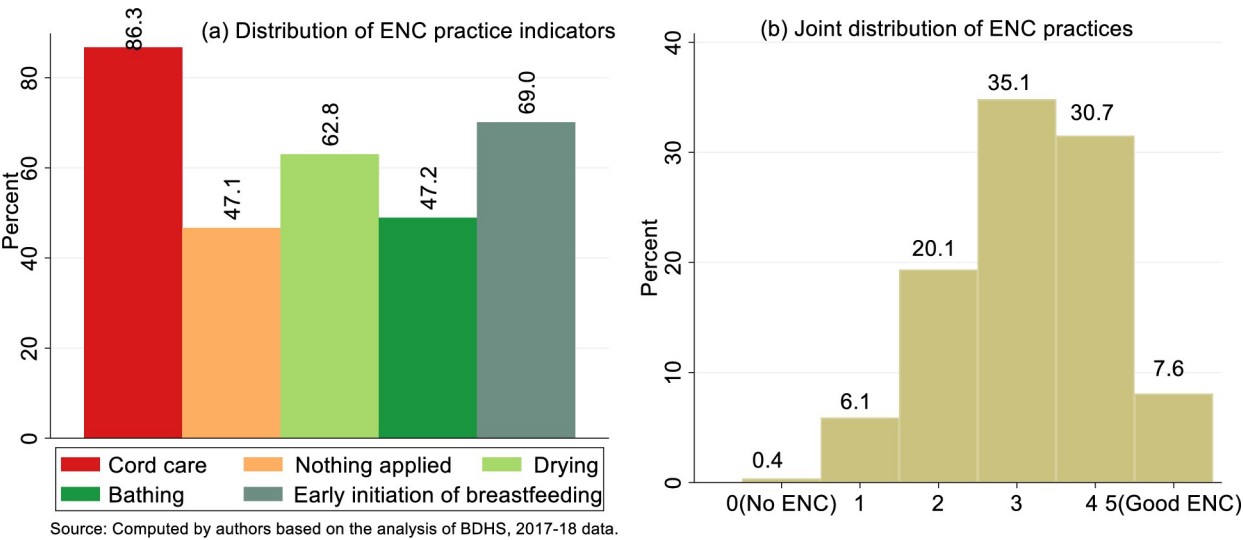

**Fig 2. Distributions of ENC practices for home deliveriesin Bangladesh, 2017–18.**

SANC or good ENC practice, an intra-class correlation (ICC) statistic was estimated in the study. The Stata command '*melogit*' was used to estimate the path coefficients of the predictors for both SANC and ENC practice. The product of two path coefficients obtained one from MEI to SANC (say, *a*) in first equation and another from SANC to ENC practice (say, *b*) in the second equation produce the amount of indirect effect of MEI on ENC practice via SANC visits (Fig 1). Again, the path coefficient from MEI to ENC practice (say, *c*) gives the direct effect of MEI on ENC practice. To draw a valid conclusion on indirect and total effects, the bias-corrected standard errors obtained from a bootstrapping sampling with 10,000 iterations were estimated. The total effect is the sum of the direct and indirect effects (i.e. total effect = *ab+c*) if the indirect effect (*ab*) is significant, otherwise the total effect is the direct effect of MEI on ENC practice. For evaluating the indirect effect, a 5% level of significance was considered in the study. Note that the covariates having p-value<0.30 were considered in the structural equation model (SEM). A likelihood ratio test (LRT) statistic was applied for checking the goodness of fit of the models.

## Results

### Descriptive statistics

The joint, as well as the independent distributions of the different components of essential newborn care practices, was presented in Fig 2. It has been observed from Fig 2A that the prevalence of cord care practice was highest (86.3%), whereas the lowest (47.1%) was observed for nothing applied to the cord. About 47% infants received delayed bathing care, about 63% of the children were dried instantly after the birth, and 69% were breastfed immediately within one hour of the birth. Fig 2B revealed that a very few newborns (0.40%) did receive none of the essential newborn cares and only 7.60% had all of the ENC practices. Of the newborns, 30.7%, 35.1%, 20.1% and 6.1% obtained four, three, two, and one ENC components, respectively. In Bangladesh, almost every newborn receives at least one ENC practice, however most babies do not receive all of the ENC practices. Babies born at home do not have access to the same services as those delivered in a hospital. As most home births take place in rural areas, family members contact an unskilled birth attendant during the delivery, where the birth attendants

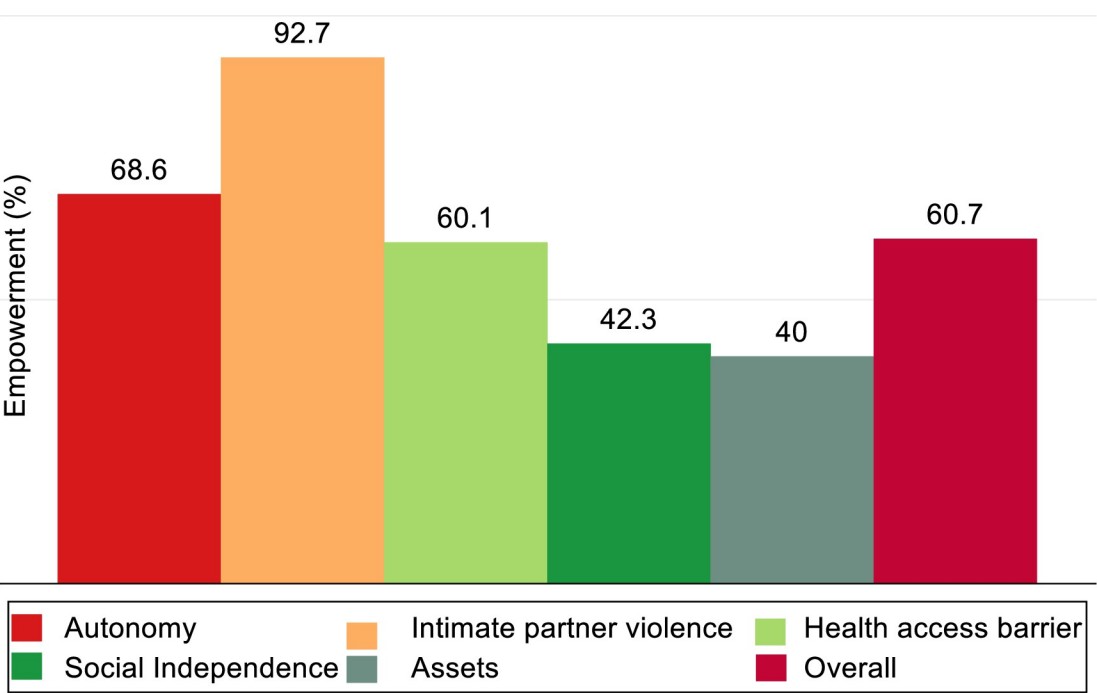

Source: Computed by authors based on the analysis of BDHS, 2017-18 data.

**Fig 3. Percentage of average empowerment in the dimensions along with overall empowerment among Bangladeshi women.**

are usually uneducated and do not follow the WHO recommended guidelines. Again, family members in rural areas are unaware of proper ENC practice, and they usually adhere to religious and cultural beliefs. As a result, the newborn does not receive sufficient care during their first few days of life.

Fig 3 illustrates the level of multidimensional and overall empowerment of Bangladeshi women. In the case of household decision-making autonomy, on average, a woman was 68.6% empowered, which implies she had participated more than two-thirds of the household's decisions. Moreover, a woman felt a positive attitude against intimate partner violence, as, on average, 92.7% of the cases, she does not support any kind of oppression from her partner. Approximately 60% of the scores related to health access barriers were covered by a mother; however, less than half of the empowerment scores regarding social independence (42.3%) and household assets (40.0%) were practiced by her, respectively. Congregating all of the dimensions along with the indicators of the empowerment index, a mother was expected to be 60.7% empowered.

Table 2 reported the results obtained from exploratory data analysis on selected covariates. It was noticed that the rate of good ENC practice, in general, was very low among the mothers who gave birth at home. More than half (54.65%) of women were average empowered, 21.63% were highly empowered, and 23.72% belonged to the low empowerment category. The prevalence of receiving at least eight antenatal care visits provided by skilled personnel was very low (5.28%) among the mothers who delivered at home in Bangladesh. Most of the women belonged to the middle age (60.58%) group and resided in rural (80.71%) area. About 53% of the selected children are either second or third child of their parents. Around two-fifths of the mothers were from the central region, one-third from the eastern region, and 29.09% from the western region of Bangladesh.

**Table 2. Percentage distribution of covariates and prevalence of good ENC for home births in Bangladesh.**

| Variables | Frequency | % | ªGood ENC Practice | ᵇp-value |
|---|---|---|---|---|
| ᶜWomen's Empowerment Index | | | | 0.003 |
| Low | 588 | 23.72 | 7.0 [4.5–9.5] | |
| Average | 1355 | 54.65 | 6.9 [5.4–8.4] | |
| High | 536 | 21.63 | 10.0 [7.3–12.8] | |
| Skilled ANC | | | | 0.011 |
| No | 2305 | 94.72 | 7.2 [6.1–8.4] | |
| Yes | 136 | 5.28 | 15.0 [9.1–21.0] | |
| Women's Current Age | | | | 0.265 |
| 15–20 | 428 | 17.27 | 6.0 [3.7–8.4] | |
| 20–30 | 1502 | 60.58 | 8.0 [6.5–9.5] | |
| 30+ | 549 | 22.15 | 7.7 [5.5–10.0] | |
| Birth Order | | | | 0.230 |
| First | 730 | 29.44 | 6.7 [4.8–8.5] | |
| Second/Third | 1305 | 52.63 | 8.5 [6.9–10.2] | |
| Fourth/Higher | 444 | 17.93 | 6.4 [3.8–9.1] | |
| Place of Residence | | | | 0.123 |
| Rural | 2001 | 80.71 | 8.1 [6.8–9.4] | |
| Urban | 478 | 29.29 | 5.7 [3.6–7.7] | |
| Region | | | | 0.008 |
| Eastern | 796 | 32.10 | 7.2 [5.3–9.1] | |
| Central | 962 | 38.81 | 5.6 [3.9–7.2] | |
| Western | 721 | 29.09 | 10.8 [8.3–13.2] | |
| **Total** | **n = 2479** | | **7.6 [6.5–8.7]** | |

*Source*: Computed by authors based on the analysis of BDHS, 2017–18 data.

*Note*. ªPrevalence of good ENC practice with 95% confidence interval (CI) by selected covariates.

ᵇp-values were calculated from chi-square test statistic to examine the association of the covariates with good ENC practice. ᶜEmpowerment: Low (score<0.50), moderate (0.50≤score<0.75), and high (score≥0.75).

The prevalence of Good ENC Practice by selected socio-economic and demographic covariates was also presented in Table 2. Women's empowerment index significantly associated (p-value = 0.003) with the good ENC practice as among highly empowered mothers 10.0% practiced good ENC to their newborns. Moreover, only 6.9% of children of average empowered and 7.0% from low empowered mothers received good ENC. Skilled antenatal care (SANC) visits also had a significant impact (p-value = 0.011) on good ENC practice. The rate of good ENC practice was 15.0% for the mothers who had at least eight ANC visits by skilled personnel, whereas 7.2% of newborns received good ENC, but their mothers did not have at least eight ANC from skilled attendants.

Association between multidimensional empowerment index (MEI) and eight or more skilled ANC visits was illustrated graphically in Fig 4A. This revealed that the rate of receiving SANC increases with the increase in MEI level. For example, for the lower MEI the prevalence of SANC was 2.27%, whereas it was 4.71% and 10.00% for average and higher MEI, respectively. Fig 4B represented the distribution of good ENC by SANC and MEI. It was found that among the mothers with lower MEI, the prevalence of good ENC practice was lower for 8 + SANC visits compared to less than 8 SANC visits. However, for the mothers with average or higher MEI the prevalence of good ENC was larger for who visited SANC compared to who did not. In general, the practice of good ENC was found to be highest among mothers who belonged to higher MEI and receiving eight or more skilled ANC.

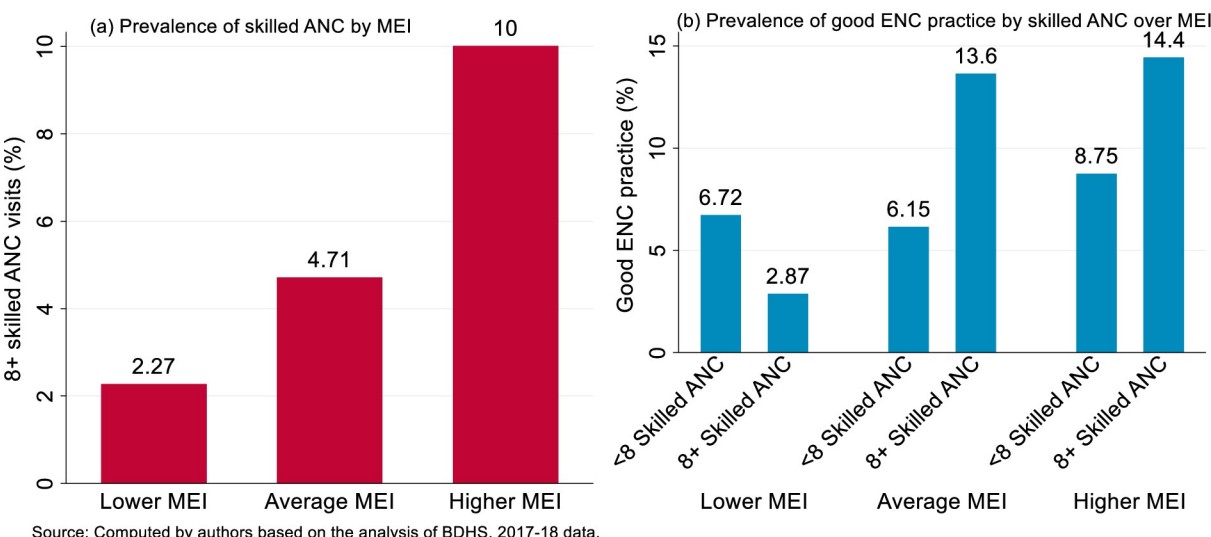

**Fig 4.** (A) Prevalence of skilled ANC visits by women's multidimensional empowerment index (MEI), and (B) the prevalence of getting good ENC practice by skilled ANC visits over MEI for the home births.

## Mediation results

Results obtained from mediation analysis considering structural equations after controlling women's current age, birth order, place of residence and region are given in Table 3. The results explained the direct effect, mediated effect via SANC, and total effect on good ENC practice. In this analysis, two multilevel binary logistic regression models were fitted: one for skilled ANC and another for good ENC practice to address the cluster effects on the outcome variables. The ICC estimates presented in Table 3 showed that skilled ANC and good ENC practices significantly varied across the clusters. The likelihood ratio test statistics reported in Table 3 also indicated that both multilevel models fitted the data well.

It was found that women's MEI was significantly associated with skilled ANC, and skilled ANC was also associated with good ENC practice. The likelihood of receiving eight or more antenatal care increases with the level of empowerment. Mothers with average and high empowerment had 108% (OR 2.08, 95% CI: 1.14–3.82) and 357% (OR 4.57, 95% CI: 2.38–8.78) higher odds of receiving SANC, respectively, compared to low empowered mothers. Moreover, the odds of good ENC practice was 106% more among mothers receiving at least eight ANC from skilled health personnel compared to their counterpart (OR 2.06, 95% CI: 1.20–3.52).

Significantly direct influence of women's empowerment was observed for good ENC practice, and mothers having high empowerment had 71% more odds of practicing good ENC compared to low empowered mothers (OR 1.71, 95% CI: 1.08–2.68). No direct and indirect association was observed between receiving good ENC practice and average empowerment as well as low empowerment. The mediating effect of highly empowered mothers through skilled ANC was significant at 5% level of significance, and among these mothers the indirect odds of receiving good ENC was 2.41 times as likely as that odds for low empowered mothers. Hence, among the highly empowered mothers, the total odds of practicing good ENC was 405% (OR 5.05, 95% CI: 1.79–14.28) higher compared to the mothers having low empowerment.

It was also found that for the women belonging to high MEI level, the indirect effect on good ENC practice was more than direct effect (67.28% versus 32.72%).

**Table 3. Mediation results obtained from structural equation modeling using BDHS, 2017–18 data.**

| Mediation | Variables | @Skilled ANC | | @Good ENC Practice | |
|---|---|---|---|---|---|
| | | β (SE) | OR [95% CI] | β (SE) | OR [95% CI] |
| **Direct Effect (DE)** | **Women's Empowerment Index** | | | | |
| | Low | *Ref.* | 1.00 | *Ref.* | 1.00 |
| | Average | 0.73 (0.31) | 2.08 [1.14–3.82]* | -0.01 (0.20) | 0.99 [0.66–1.47] |
| | High | 1.52 (0.33) | 4.57 [2.38–8.78]*** | 0.53 (0.23) | 1.71 [1.08–2.68]* |
| | **Skilled ANC** | | | | |
| | No | | | *Ref.* | 1.00 |
| | Yes | | | 0.72 (0.27) | 2.06 [1.20–3.52]** |
| **Mediated Effect (ME) *via* Skilled ANC** | **Women's Empowerment Index** | | | | |
| | Low | | | *Ref.* | 1.00 |
| | Average | | | 0.53 (0.30) | 1.54 [0.95–3.02] |
| | High | | | 1.09 (0.47) | 2.41 [1.18–7.57]* |
| **Total Effect (TE)** | **Women's Empowerment Index** | | | | |
| | Low | | | *Ref.* | 1.00 |
| | Average | | | 0.53 (0.30) | 1.54 [0.95–3.02] |
| | High | | | 1.62 (0.53) | 5.05 [1.79–14.28]** |
| **Proportion of Mediated Effect (PME)** $[PME = \frac{ME}{TE} \times 100]$ | **Women's Empowerment Index** | | | | |
| | Low | | | | *Ref.* |
| | Average | | | | - |
| | High | | | | 67.28% |
| **LR Test Statistic** | | **59.91***** | | **31.85**** | |
| **Intra-class correlation (ICC)** | | **0.26 [0.14–0.43]***** | | **0.13 [0.05–0.27]**** | |

*Source*: Computed by authors based on the analysis of BDHS, 2017–18 data.

*Note.* @Adjusted by current age, birth order, place of residence and region.

*p<0.05.

**p<0.01

***p<0.001. β regression coefficients, SE standard error, OR odds ratio, Ref. reference category.

LR likelihood ratio.

## Discussion

The study explored the scenario of good ENC practice among the mothers to their newborns born at home in Bangladesh. Only 7.6% of mothers followed good ENC practice and independently, 86.3% used clean and boiled instruments to cut the umbilical cord, 47.1% did not apply anything to the cord, 62.8% dried the newborn in 5 minutes, 47.2% bathed the baby 72 hours or later after birth and 69.0% initiated early breastfeeding within 1 hour of birth. Although the prevalence of most of the ENC components are in line or close with other studies conducted in Ethiopia [62–67], the prevalence of overall ENC practice is very low in Bangladesh, and it is far away from the target of 4th HPNSP (Health, Population and Nutrition Sector Program) to achieve the prevalence of 25% by 2022 [4,12,24]. No significant improvement was observed in good ENC practice in Bangladesh over the previous years. For example, in 2011, the prevalence was around 2% and it was increased to 6% in 2014 [68,69]. But this prevalence was much higher in Ethiopia (38%) [8,52], Sudan (58.9%) [70], Eastern Tigray (92.9%) [64] and Uganda (11.7%) [50]. The ENC practice was also much higher in South Asian countries such as Nepal (70.7%) [51] and India (66.7%) [71] compared to Bangladesh. This may happen because of the better health system and awareness programs on ENC practice for the good health of newborns organized by various healthcare organizations in the study area [8]. Therefore,

Bangladesh needs to revise policy-making interventions for accelerating the rate of good ENC practice.

The results obtained from the present study can contribute to policymaking to revise some interventions. In the study, a mediation technique was applied to observe the direct, indirect and total effects of women's empowerment on good ENC practice through the SANC. Both path coefficients of MEI to SANC and SANC to ENC practice were found to be significant in this study which was used to compute the indirect effect. Note that the hypothesis regarding indirect effect as well as the total effect was tested by using a bias-corrected standard error estimate obtained from 10,000 bootstrapping samples.

The study revealed that both women's multidimensional empowerment index (MEI) and skilled antenatal care (SANC) had shown the significant association with good ENC practice in both adjusted and unadjusted analyses. The mothers who received eight or more antenatal care visits from skilled health professionals during their pregnancy period were 85% more likely to practice good ENC compared to their counterparts. Similar finding was found in other studies conducted in Bangladesh, Southern Ethiopia, Northern Ghana, Eastern Uganda, Sindhuli District of Nepal, Northwest Ethiopia [49,50,61,62,72]. This is because the health professionals are generally educated and follow national and international quality policies and regulations regarding maternal and child health care and hence pregnant mothers taking antenatal care from skilled health workers can acquire knowledge about newborn health care practice [53,73–75]. Moreover, mothers having more follow-ups during pregnancy period may have more access to the information regarding maternal and child health obtained from healthcare providers.

The main exposure of this study was the MEI which was assessed through several dimensions as well as indicators related to women's empowerment. The study revealed that women's empowerment had significant positive effect on skilled antenatal care. The likelihood of receiving 8 or more SANC increased as the level of empowerment increased. The finding was consistent with another study conducted in Southeast Asia and the study revealed that the ANC utilization and general health status can be improved by enhancing women's health related empowerment and social status [48,76]. Another study conducted in Bangladesh also found the similar finding [46].

The direct and indirect effects of average empowerment on ENC practice were not found as significant, but for high empowerment, these effects on good ENC practice were found to be significant, and so did the total effect. Therefore, the study explored that highly empowered mothers were more likely to practice good ENC compared to low empowered mothers (OR 5.05), having the odds ratio of 1.71 from direct influence and 2.41 from indirect influence via SANC. Recall that the highly empowered women satisfied at least 75% of the selected empowerment indicators, which implies that highly empowered women have more access to take their family decisions like healthcare, and also have control over health resources that improve their health as well as the health of their children [40–42,44,45,77]. Moreover, the empowered women have more chance to be educated, employed and economically solvent that confirmed the potential role in maternal and child health by utilizing the maternal healthcare knowledge [78–80]. Access to medical facilities is also undermined among the lower empowered women, which in turn may create barriers in accessing child healthcare. In addition, highly empowered mothers are more likely to follow the guidelines or advice given by healthcare providers during pregnancy or through media such as newspapers/television, which in turn influences mothers to practice good ENC for the good health of newborns as well as mothers.

Interestingly, the indirect effect of high empowerment through skilled antenatal care was greater than its direct effect on good ENC practice. For example, a highly empowered mother received good ENC practice through SANC indirectly with around 67% and 33% directly. This

may be because highly empowered women receiving eight or more antenatal care from skilled health professionals have been able to access maternal and child health services. On the other hand, the empowered mothers may be unable to practice good ENC properly if they fail to receive skilled health care during pregnancy. Therefore, the study revealed that the skilled antenatal care plays an important mediating role to strengthen the relationship between women's empowerment and good ENC practice.

## Strength and limitations

This is the first study that used multidimensional empowerment index (MEI) as a potential contributor in practicing good ENC. The MEI was developed as a composite index assessing five dimensions associated with women's empowerment. In the study, a mediation technique was applied considering structural equation modeling to find out the mediating role of skilled antenatal care on the relationship between women's empowerment and good ENC practice. The eight or more antenatal care received from skilled health professionals was defined as skilled antenatal care (SANC) following WHO recommendation. This study found the significant mediating effect as well as the direct effect of women's empowerment on good ENC practice. Therefore, it can be used as a yardstick and motivation for further study in related areas.

The study also has some limitations. There may have been recalling bias in the study as the mothers selected for the study might failed to remember what they did for their newborns in the early neonatal period. Second, it may not be possible to establish the causal relationship between MEI and good ENC practice as the design of the study was cross-sectional. Moreover, only home births born preceding three years of the survey were considered.

## Conclusion

The study evaluated good ENC practice through a number of neonatal care practices, such as safe cord care, optimal thermal care, and neonatal feeding. We have also explored the current rates of skilled antenatal care and women's empowerment in Bangladesh. The study indicated that the prevalence of good ENC practice in the country was very low which could be accelerated by both skilled antenatal care and women's empowerment, where skilled antenatal care plays an important mediating role in improving good ENC practice among highly empowered mothers. The study recommends that a woman should be highly empowered if she possesses, on an average, not less than 75% of empowerment in terms of family decision, attitude towards intimate partner violence, social independence, access to healthcare and economic status. Since education is an important indicator to become empowered, women should be highly educated so that they can contribute in breaking barriers to healthcare access, domestic violence, in making household decisions, and in becoming socially independent, and hence women will be multidimensionally empowered. Therefore, policymakers should emphasize in education especially in primary or secondary school so that girls can understand the importance of women's empowerment. The study further argues that the woman should follow the recent guidelines recommended by WHO for antenatal care follow-up from skilled health workers for good ENC practice. A revised intervention is also needed in receiving skilled antenatal care follow ups especially in rural areas.

## Acknowledgments

We would like to thank DHS MEASURE for allowing us to use the data set. We would also like to thank the National Institute of Population Research and Training (NIPORT) for implementing BDHS, 2017–2018.

## Author Contributions

**Conceptualization:** Kanchan Kumar Sen, Wasimul Bari.

**Data curation:** Kanchan Kumar Sen, Ahsan Rahman Jamee.

**Formal analysis:** Kanchan Kumar Sen.

**Methodology:** Kanchan Kumar Sen, Ahsan Rahman Jamee.

**Supervision:** Wasimul Bari.

**Writing – original draft:** Kanchan Kumar Sen, Ahsan Rahman Jamee.

**Writing – review & editing:** Wasimul Bari.

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
