## [Decision Letter · Decision Letter 0]

14 Nov 2022

PONE-D-22-01899Women's multidimensional empowerment index and essential newborn care practice in Bangladesh: the mediating role of skilled antenatal care follow-upsPLOS ONE

Dear Dr. Kanchan Kumar Sen, 

Thank you for submitting your manuscript to PLOS ONE. After careful consideration, we feel that it has merit but does not fully meet PLOS ONE’s publication criteria as it currently stands. Therefore, we invite you to submit a revised version of the manuscript that addresses the points raised during the review process.

We look forward to receiving your revised manuscript.

Kind regards,

Gouranga Lal Dasvarma, PhD

Academic Editor

PLOS ONE

Journal Requirements:

2. In the ethics statement in the Methods and online submission information, please ensure that you have specified (1) whether consent was informed and (2) what type you obtained (for instance, written or verbal, and if verbal, how it was documented and witnessed). If your study included minors, state whether you obtained consent from parents or guardians. If the need for consent was waived by the ethics committee, please include this information.

Additional Editor Comments (if provided):

Under the provisions of PLOS One Editorial Office, I have invited myself as the second reviewer of your manuscript.

Reviewers' comments:

Reviewer's Responses to Questions

**Comments to the Author**

1. Is the manuscript technically sound, and do the data support the conclusions?

Reviewer #1: Yes

Reviewer #2: Yes

2. Has the statistical analysis been performed appropriately and rigorously? 

Reviewer #1: Yes

Reviewer #2: Yes

3. Have the authors made all data underlying the findings in their manuscript fully available?

Reviewer #1: Yes

Reviewer #2: Yes

4. Is the manuscript presented in an intelligible fashion and written in standard English?

Reviewer #1: Yes

Reviewer #2: No

5. Review Comments to the Author

Reviewer #1: Thank you for the opportunity to review this manuscript. The topic is important for public health issue, especially in maternal and child health. I have some comments to improve the manuscript:

Introduction: The topic of empowerment has to be introduced in this section. How is women empowerment defined? Is there a framework used to define women’s empowerment components? This must be justified and supported by literature.

Methods: Explain why author used weighting approach to create the empowerment component.

Results: In Table 2, column 4 is not clear.

Discussion:

There could be more a more detailed comparison of the results here with previous results. As we go from the top quartile of empowerment to the bottom quartile, how large is the change in the probability of SANC or ENC in other studies? Even if the other studies use different methods and different empowerment measures, it should still be possible to the infer the differences between the top and bottom quartiles in these studies.

Policy implications are not clear.

Conclusions drawn are supported by findings.

Reviewer #2: This is a useful and innovative study with potential policy implications. It is not certain to this reviewer whether the idea of using antenatal care-visits as a mediating factor between women’s empowerment and essential newborn care is completely new, but it is a refreshing approach, especially in measuring women’s empowerment from a multi-dimensional angle. However, there are several points, stated below, which need to be addressed satisfactorily before the manuscript may be considered for publication.

General comments:

• Language editing: The manuscript must be professionally edited for English, particularly for use of articles such as “the” and tense. The manuscript is supposed to describe and explain the findings of a current research, yet some of the sentences read as if they are reporting past research.

• Writing style. The style of writing is often too cryptic, especially in the presentation of tables. Table and figure headings should be self-explanatory, and the source of all the tables and figures must be noted. Even if the tables and figures are drawn by the authors based on their analysis of data, these should be mentioned as such.

• Statistical analyses of data: In general, the idea of calculating the direct and indirect effects of women’s empowerment on ENC practice is noteworthy. However, please confirm whether it was necessary to adjust for complex design in conducting logistic regression on data extracted from BDHS 2017-18, which is based on complex sampling. Please also explain what was involved in “wrangling” the data, as BDHS 2017-18 data must have been cleaned before being released for use.

Specific comments:

• Lines 14-15, and 138-139. Please explain why only home births are considered in this research. According to Bangladesh Demographic and Health Survey 2017-18 (BDHS 2017-18. Chapter 9), almost one half (49%) of the births in the preceding three years were delivered at a health facility.

• Line 58. “intrapartum, conditions of labour”. Do you mean intrapartum conditions of labour (i.e. no comma)?

• Lines 142-154. Re initial care for babies after birth: Does “cord care” include Used safe

delivery kit/bag or boiled blade during delivery, Applied chlorhexidine after umbilical

cord was cut and tied (BDHS 2017-18 report Table 9.24? Re joint groupings. Where have you shown in your manuscript the joint groupings for safe cord care and optimal thermal care respectively, reported from previous studies?

• Lines 156-162. Re components of women’s empowerment index. Why is women’s education used as a barrier to accessing healthcare and not as a part of household decision making autonomy? How valid is the assumption to consider household assets as proxy for women’s assets, when in fact household assets are most likely to be held jointly with husbands and possibly members of the extended family?

• Lines 162-166. What is the justification of assigning equal weights to each dimension? Why did you not use Factor Analysis and find the weights from Factor Loadings?

• Lines 177-185. Formula for individual score. Please re-write the formula as it does not contain the symbol ‘n’ (noted in Line 180) and symbols denoting the weights of indicator and dimension.

• Lines 191-196. Re the new WHO guidelines about at least eight Skilled ANC (SANC) visits. The new guidelines were formulated in 2016. In your analysis you have applied the new recommendations to births and pregnancies occurring in the three years preceding the survey, which was conducted in 2017-18, i.e. births and pregnancies occurring during 2015-2018 or before. As such, many births and associated SANC visits (or non-visits) must have happened before the new guidelines came into effect (when at least 4 SANC visits were recommended). Under such circumstances how justified are you in considering a cut-off of less than 8 and 8 or more SANC visits in analysing the mediating effects of skilled antenatal care?

• Line 198. “gender of index child”. It appears from an inspection of Table 2 that the index child is the child whose antenatal care is the subject of this study. As such, selecting “gender of the index child” as a control variable does not make sense because it is not possible to know the gender of the child at the antenatal stage, unless tests like ultrasound or amniocentesis are done, which most women would not have done in this case. Therefore, this variable should be removed from the list of control variables and the analysis should be re-done.

• Lines 207-211. Statistical Analysis. Please confirm whether it was necessary to adjust for complex design in conducting logistic regression on data extracted from BDHS 2017-18, which is based on complex sampling. Please also explain what was involved in “wrangling” the data, as BDHS 2017-18 data must have been cleaned before being released for use.

• Table 2. Please make the table heading self-explanatory by stating the covariates of which indicator you have presented in the table based on what test. Please also provide, in parentheses the range of values of Low, Average and High Women’s Empowerment Index. What values are shown under the column heading “Good ENC Practice”? Are these the point estimates of each variable and their 95% Confidence Intervals? Please write the source of this table (and other tables and figures). If this table is based on your own calculations, then please write the source somewhat as this: “Computed by the authors based on their analysis of BDHS 2017-18 data”.

• Fig 1. (Should write in full, as “Figure 1”). Please calculate if possible, the Direct Effect of MEI on ENC Practice, and the effects of MEI on Skilled ANC and Skilled ANC on ENC Practice respectively and show them in the diagram.

• Fig 2(a).This figure (and its parent table in BDHS 2017-18-Table 9.24) appears a little confusing with respect to cord care and nothing applied to cord, both of which are indicators of cord care. Therefore, please explain how 86.7% of home delivered births in the last three years had Used safe delivery kit/bag or boiled blade during delivery, applied chlorhexidine after umbilical cord was cut and tied and 44.1% had nothing applied to cord. Should these two percentages not add to 100?

• Fig 2(b). “ENC practice is considered as a binary random variable, where 1 indicates good ENC and 0 for the poor ENC. A newborn is said to be received good ENC if all five components were practiced; whereas the practice is poor if he/she missed any of the components”. (Lines 151-164 of this manuscript). In the context of this definition, please explain the five categories of ENC practice displayed in Fig 2(b).

• Please also examine all the other diagrams (and tables) and check them for missing information or lack of clarity.

• Lines 410-422. In conclusion you have recommended “that a woman should be highly empowered …”. Can you suggest what specific measures should be taken to make women highly empowered in order to possess “not less than 75% of empowerment”?

6. PLOS authors have the option to publish the peer review history of their article (what does this mean?). If published, this will include your full peer review and any attached files.

Reviewer #1: No

Reviewer #2: No

---

## [Author Response · Author response to Decision Letter 0]

4 Dec 2022

Authors’ Response to Journal Requirements and Reviewer’s Comments

Journal Requirements

Comment # 01

Authors’ Response:

The PLOS ONE's style requirements have been followed in writing the manuscript.

Comment # 02

In the ethics statement in the Methods and online submission information, please ensure that you have specified (1) whether consent was informed and (2) what type you obtained (for instance, written or verbal, and if verbal, how it was documented and witnessed). If your study included minors, state whether you obtained consent from parents or guardians. If the need for consent was waived by the ethics committee, please include this information.

Authors’ Response:

It has been taken care of.

Comment # 03

Your ethics statement should only appear in the Methods section of your manuscript. If your ethics statement is written in any section besides the Methods, please move it to the Methods section and delete it from any other section. Please ensure that your ethics statement is included in your manuscript, as the ethics statement entered into the online submission form will not be published alongside your manuscript.

Authors’ Response:

The ethics statement has been moved to the Methods section in the revised manuscript.

Reviewer Comments

Reviewer # 1

Thank you for the opportunity to review this manuscript. The topic is important for public health issue, especially in maternal and child health. I have some comments to improve the manuscript:

Dear Reviewer:

Thank you very much for your effort in reviewing our manuscript and for your thoughtful suggestions. We have modified the manuscript according to your comment and detailed corrections are listed below, point by point. All changes are highlighted in blue color in this revised manuscript with track changes. 

Comment # 01

Introduction: The topic of empowerment has to be introduced in this section. How is women empowerment defined? Is there a framework used to define women’s empowerment components? This must be justified and supported by literature.

Authors’ Response:

Thank you very much for your valuable comment. We have mentioned the women’s empowerment along with its dimensions in the introduction section of the revised manuscript (Please see the lines 91-100 of the revised manuscript file). 

Comment # 02

Methods: Explain why author used weighting approach to create the empowerment component.

Authors’ Response:

The previous studies constructed multiple components or factors of women’s empowerment. But we constructed single composite score to define the level of women’s empowerment using a weighting approach. To construct the weighting score, we followed a study on multidimensional poverty measures proposed by Alkire and Foster (2011). The reason has been in the method section. (Please see the lines 203-207)

Reference. Alkire, S., & Foster, J. (2011). Counting and multidimensional poverty measurement. Journal of public economics, 95(7-8), 476-487. 

Comment # 03

Results: In Table 2, column 4 is not clear.

Authors’ Response:

We revised the Table 2, and we included a footnote in the table for understanding the table.

Comment # 04

Discussion:

There could be more a more detailed comparison of the results here with previous results. As we go from the top quartile of empowerment to the bottom quartile, how large is the change in the probability of SANC or ENC in other studies? Even if the other studies use different methods and different empowerment measures, it should still be possible to the infer the differences between the top and bottom quartiles in these studies. 

Authors’ Response:

Thank you very much for your comment. To the best of our knowledge, no research has been conducted on the relationship between women's empowerment and ENC practice. As a result, we were unable to compare our findings to those of other studies.

Comment # 05

Policy implications are not clear. Conclusions drawn are supported by findings.

Authors’ Response:

We revised the conclusion part in the revised manuscript.

Reviewer # 2

This is a useful and innovative study with potential policy implications. It is not certain to this reviewer whether the idea of using antenatal care-visits as a mediating factor between women’s empowerment and essential newborn care is completely new, but it is a refreshing approach, especially in measuring women’s empowerment from a multi-dimensional angle. However, there are several points, stated below, which need to be addressed satisfactorily before the manuscript may be considered for publication.

Dear Reviewer:

Thank you very much for your effort in reviewing our manuscript and for your valuable comments. We have modified the manuscript according to your comment and detailed corrections are listed below, point by point. All changes are highlighted in blue color in this revised manuscript with track changes. 

Comment # 01

Language editing: The manuscript must be professionally edited for English, particularly for use of articles such as “the” and tense. The manuscript is supposed to describe and explain the findings of a current research, yet some of the sentences read as if they are reporting past research.

Authors’ Response:

Thank you very much. The manuscript has been revised accordingly.

Comment # 02

Writing style. The style of writing is often too cryptic, especially in the presentation of tables. Table and figure headings should be self-explanatory, and the source of all the tables and figures must be noted. Even if the tables and figures are drawn by the authors based on their analysis of data, these should be mentioned as such.

Authors’ Response:

It has been taken care of.

Comment # 03

Statistical analyses of data: In general, the idea of calculating the direct and indirect effects of women’s empowerment on ENC practice is noteworthy. However, please confirm whether it was necessary to adjust for complex design in conducting logistic regression on data extracted from BDHS 2017-18, which is based on complex sampling. Please also explain what was involved in “wrangling” the data, as BDHS 2017-18 data must have been cleaned before being released for use.

Authors’ Response:

Thank you very much for your valuable comments. BDHS data followed a complex survey design. In our revised manuscript, we revised our all analysis after addressing complex survey design using svy setting in Stata. We also took into account the cluster variation in our regression analysis using multilevel or mixed-effect binary logistic regression model. Again, the word “wrangling” was used in the statistical analysis section. Actually, in our selected dataset, some characteristics were missing for some individuals, and hence we discarded these individuals from our dataset. For this case, we used the mentioned word “wrangling”. In the revised manuscript, we delated the word.

(Specific Comments) 

Comment # 04

Lines 14-15, and 138-139. Please explain why only home births are considered in this research. According to Bangladesh Demographic and Health Survey 2017-18 (BDHS 2017-18. Chapter 9), almost one half (49%) of the births in the preceding three years were delivered at a health facility.

Authors’ Response:

The Bangladesh demographic and health survey (BDHS), 2017-18 reported the essential newborn care practices for home births only, In the original data set, we also did not find the information on ENC indicators for institutional births. For this reason, we only included home births only in our study. We added this reason in the revised manuscript (Please see the lines: 152-154).

Comment # 05

Line 58. “intrapartum, conditions of labour”. Do you mean intrapartum conditions of labour (i.e. no comma)?

Authors’ Response:

We have written the words correctly on the manuscript.

Comment # 06

Lines 142-154. Re initial care for babies after birth: Does “cord care” include Used safe

delivery kit/bag or boiled blade during delivery, Applied chlorhexidine after umbilical

cord was cut and tied (BDHS 2017-18 report Table 9.24? Re joint groupings. Where have you shown in your manuscript the joint groupings for safe cord care and optimal thermal care respectively, reported from previous studies?

Authors’ Response:

We revised the lines 142-154. According to BDHS 2017-18 report cord care include used safe delivery kit/bag or boiled blade during delivery. The nothing applied to the cord or only chlorhexidine applied is another component of ENC.

Comment # 07

Lines 156-162. Re components of women’s empowerment index. Why is women’s education used as a barrier to accessing healthcare and not as a part of household decision making autonomy? How valid is the assumption to consider household assets as proxy for women’s assets, when in fact household assets are most likely to be held jointly with husbands and possibly members of the extended family?

Authors’ Response:

It has been taken care of. Women’s education is included in the social independence domain of women’s empowerment. 

Comment # 08

Lines 162-166. What is the justification of assigning equal weights to each dimension? Why did you not use Factor Analysis and find the weights from Factor Loadings?

Authors’ Response:

The previous studies constructed multiple components or factors of women’s empowerment. But we constructed single composite score to define the level of women’s empowerment using a weighting approach. To construct the weighting score, we followed a study on multidimensional poverty measures proposed by Alkire and Foster (2011). The reason has been in the method section. (Please see the lines 203-207)

Reference: Alkire, S., & Foster, J. (2011). Counting and multidimensional poverty measurement. Journal of public economics, 95(7-8), 476-487.

Comment # 09

Lines 177-185. Formula for individual score. Please re-write the formula as it does not contain the symbol ‘n’ (noted in Line 180) and symbols denoting the weights of indicator and dimension.

Authors’ Response:

It has been corrected.

Comment # 10

Lines 191-196. Re the new WHO guidelines about at least eight Skilled ANC (SANC) visits. The new guidelines were formulated in 2016. In your analysis you have applied the new recommendations to births and pregnancies occurring in the three years preceding the survey, which was conducted in 2017-18, i.e. births and pregnancies occurring during 2015-2018 or before. As such, many births and associated SANC visits (or non-visits) must have happened before the new guidelines came into effect (when at least 4 SANC visits were recommended). Under such circumstances how justified are you in considering a cut-off of less than 8 and 8 or more SANC visits in analysing the mediating effects of skilled antenatal care?

Authors’ Response:

One of the objectives of this study is to examine whether ENC practice increases with the increase of ANC visits. For this reason, we have used the newly proposed cut off point for ANC visits to observe how it works on ENC practices. The reason has also been included in the revised manuscript.

Comment # 11

Line 198. “gender of index child”. It appears from an inspection of Table 2 that the index child is the child whose antenatal care is the subject of this study. As such, selecting “gender of the index child” as a control variable does not make sense because it is not possible to know the gender of the child at the antenatal stage, unless tests like ultrasound or amniocentesis are done, which most women would not have done in this case. Therefore, this variable should be removed from the list of control variables and the analysis should be re-done.

Authors’ Response:

We reanalyzed the data after discarding the variable “gender of index child”.

Comment # 12

Lines 207-211. Statistical Analysis. Please confirm whether it was necessary to adjust for complex design in conducting logistic regression on data extracted from BDHS 2017-18, which is based on complex sampling. Please also explain what was involved in “wrangling” the data, as BDHS 2017-18 data must have been cleaned before being released for use.

Authors’ Response:

BDHS data followed a complex survey design. In our revised manuscript, we revised our all analysis after addressing complex survey design using svy setting in Stata. We also took into account the cluster variation in our regression analysis using multilevel or mixed-effect binary logistic regression model. Again, the word “wrangling” was used in the statistical analysis section. Actually, in our selected dataset, some characteristics were missing for some individuals, and hence we discarded these individuals from our dataset. For this case, we used the mentioned word “wrangling”. In the revised manuscript, we delated the word.

Comment # 13

Table 2. Please make the table heading self-explanatory by stating the covariates of which indicator you have presented in the table based on what test. Please also provide, in parentheses the range of values of Low, Average and High Women’s Empowerment Index. What values are shown under the column heading “Good ENC Practice”? Are these the point estimates of each variable and their 95% Confidence Intervals? Please write the source of this table (and other tables and figures). If this table is based on your own calculations, then please write the source somewhat as this: “Computed by the authors based on their analysis of BDHS 2017-18 data”.

Authors’ Response:

It has been taken care of.

Comment # 14

Fig 1. (Should write in full, as “Figure 1”). Please calculate if possible, the Direct Effect of MEI on ENC Practice, and the effects of MEI on Skilled ANC and Skilled ANC on ENC Practice respectively and show them in the diagram.

Authors’ Response:

The Direct Effect of MEI on ENC Practice, and the effects of MEI on Skilled ANC and Skilled ANC on ENC practice were reported in Table 3.

Comment # 15

Fig 2(a).This figure (and its parent table in BDHS 2017-18-Table 9.24) appears a little confusing with respect to cord care and nothing applied to cord, both of which are indicators of cord care. Therefore, please explain how 86.7% of home delivered births in the last three years had Used safe delivery kit/bag or boiled blade during delivery, applied chlorhexidine after umbilical cord was cut and tied and 44.1% had nothing applied to cord. Should these two percentages not add to 100?

Authors’ Response:

According to BDHS 2017-18 report we measure the five ENC components. Cord care include used safe delivery kit/bag or boiled blade during delivery. The nothing applied to the cord or only chlorhexidine applied is another component of ENC. Therefore, these two percentages are not to be 100. It is noted that nothing applied to the cord means no substances are used after cutting the umbilical cord.

Comment # 16

Fig 2(b). “ENC practice is considered as a binary random variable, where 1 indicates good ENC and 0 for the poor ENC. A newborn is said to be received good ENC if all five components were practiced; whereas the practice is poor if he/she missed any of the components”. (Lines 151-164 of this manuscript). In the context of this definition, please explain the five categories of ENC practice displayed in Fig 2(b).

Authors’ Response:

It has been taken care of.

Comment # 17

Please also examine all the other diagrams (and tables) and check them for missing information or lack of clarity.

Authors’ Response:

It has been taken care of accordingly.

Comment # 18

Lines 410-422. In conclusion you have recommended “that a woman should be highly empowered …”. Can you suggest what specific measures should be taken to make women highly empowered in order to possess “not less than 75% of empowerment”?

Authors’ Response:

The conclusion has been revised accordingly.

---

## [Editor Report · Decision Letter 1]

11 Jan 2023

PONE-D-22-01899R1Women's multidimensional empowerment index and essential newborn care practice in Bangladesh: the mediating role of skilled antenatal care follow-upsPLOS ONE

Dear Dr. Kanchan Kumar Sen, 

Thank you for submitting your manuscript to PLOS ONE. After careful consideration, we feel that it has merit but does not fully meet PLOS ONE’s publication criteria as it currently stands. Therefore, we invite you to submit a revised version of the manuscript that addresses the points raised during the review process. These are only minor comments and should be able to be addressed in less than a week.

PONE-D-22-01899R1  Kanchan Kumar Sen

Reviewer#2 - Comment 10. “Lines 191-196.”Re the new WHO guidelines about at least eight Skilled ANC (SANC) visits. The new guidelines were formulated in 2016. In your analysis you have applied the new recommendations to births and pregnancies occurring in the three years preceding the survey, which was conducted in 2017-18, i.e. births and pregnancies occurring during 2015-2018 or before. As such, many births and associated SANC visits (or non-visits) must have happened before the new guidelines came into effect (when at least 4 SANC visits were recommended). Under such circumstances how justified are you in considering a cut-off of less than 8 and 8 or more SANC visits in analysing the mediating effects of skilled antenatal care?” is not addressed satisfactorily (Lines 231-240 of the revised manuscript).
**Please explain the reason why you have considered a cut-off of less than 8 and 8 or more SANC visits while the previous guideline of 4 skilled ANC visits was still followed when most of the births included in your study took place**.Reviewer#2 – Comment 16. Your response “It has ben taken care of”. How have you taken care of it? Have you removed Figure 2(b) and retained Figure 2(a) as the only Figure 2??==============================

We look forward to receiving your revised manuscript.

Kind regards,

Gouranga Lal Dasvarma, PhD

Academic Editor

PLOS ONE

Journal Requirements:

Additional Editor Comments:

Thank you for revising the manuscript. Two minor comments still remain to be addressed as follows:

PONE-D-22-01899R1 Kanchan Kumar Sen

• Reviewer#2 - Comment 10. “Lines 191-196.”Re the new WHO guidelines about at least eight Skilled ANC (SANC) visits. The new guidelines were formulated in 2016. In your analysis you have applied the new recommendations to births and pregnancies occurring in the three years

preceding the survey, which was conducted in 2017-18, i.e. births and pregnancies occurring during 2015-2018 or before. As such, many births and associated SANC visits (or non-visits) must have happened before the new guidelines came into effect (when at least 4 SANC visits

were recommended). Under such circumstances how justified are you in considering a cut-off of less than 8 and 8 or more SANC visits in analysing the mediating effects of skilled antenatal care?” is not addressed satisfactorily (Lines 231-240 of the revised manuscript). Please

explain the reason why you have considered a cut-off of less than 8 and 8 or more SANC visits while the previous guideline of 4 skilled ANC visits was still followed when most of the births included in your study took place.

• Reviewer#2 – Comment 16. Your response “It has ben taken care of”. How have you taken care of it? Have you removed Figure 2(b) and retained Figure 2(a) as the only Figure 2??

---

## [Author Response · Author response to Decision Letter 1]

15 Jan 2023

Authors’ Response to Reviewer’s Comments

Reviewer # 2

Dear Reviewer:

Thank you very much for your effort in reviewing our manuscript and for your valuable comments. We have modified the manuscript according to your comments, and detailed corrections are listed below, point by point. All changes are highlighted in blue color in this revised manuscript with track changes. 

Comment # 1

Comment 10. “Lines 191-196.”Re the new WHO guidelines about at least eight Skilled ANC (SANC) visits. The new guidelines were formulated in 2016. In your analysis you have applied the new recommendations to births and pregnancies occurring in the three years preceding the survey, which was conducted in 2017-18, i.e. births and pregnancies occurring during 2015-2018 or before. As such, many births and associated SANC visits (or non-visits) must have happened before the new guidelines came into effect (when at least 4 SANC visits were recommended). Under such circumstances how justified are you in considering a cut-off of less than 8 and 8 or more SANC visits in analyzing the mediating effects of skilled antenatal care?” is not addressed satisfactorily (Lines 231-240 of the revised manuscript). Please explain the reason why you have considered a cut-off of less than 8 and 8 or more SANC visits while the previous guideline of 4 skilled ANC visits was still followed when most of the births included in your study took place.

Authors’ Response:

Thank you for your thoughtful observation. All pregnant women and adolescent girls should follow the updated WHO recommendation released in 2016 to receive antenatal care (ANC). The guideline is also intended for national and local policymakers, implementers and managers of national and local mother and child health programs, non-governmental and other organizations, public health policymakers and health professionals. In accordance with a human rights-based perspective, the guide strives to capture the complex nature of the concerns facing ANC health care practices and delivery, and to promote person-centered health and well-being over death and morbidity prevention. As a result, adherence to the guidelines is necessary. In this study, however, we also followed the WHO guideline on ANC visits (2016) to observe how eight or more ANC visits influence good essential newborn care practice. Despite the fact that many of the births and related ANC visits (or non-visits) included in the study occurred prior to the revised guidelines, our study will provide more evidence to follow the most recent WHO guideline on ANC visits to ensure appropriate ENC practice. The study looked into whether or not the new guideline works in terms of receiving good ENC before adopting it. Furthermore, the current WHO recommendation on ANC visits was used in the literature, which included data from the 2018 Demographic and Health Survey [1–3]. The reason has also been included in the revised manuscript. The changes are highlighted in the revised manuscript with blue color (See page- 13,14, lines- 228-235).

References

1. Ahinkorah BO, Seidu AA, Budu E, Mohammed A, Adu C, Agbaglo E, et al. Factors associated with the number and timing of antenatal care visits among married women in Cameroon: Evidence from the 2018 Cameroon Demographic and Health Survey. J Biosoc Sci. 2021. doi:10.1017/S0021932021000079

2. Imo CK. Influence of women’s decision-making autonomy on antenatal care utilisation and institutional delivery services in Nigeria: evidence from the Nigeria Demographic and Health Survey 2018. BMC Pregnancy Childbirth. 2022;22: 1–12. doi:10.1186/s12884-022-04478-5

3. Tessema ZT, Tesema GA, Yazachew L. Individual-level and community-level factors associated with eight or more antenatal care contacts in sub-Saharan Africa: Evidence from 36 sub-Saharan African countries. BMJ Open. 2022;12: 1–8. doi:10.1136/bmjopen-2021-049379

Comment # 2

Comment 16. Your response “It has been taken care of”. How have you taken care of it? Have you removed Figure 2(b) and retained Figure 2(a) as the only Figure 2??

Authors’ Response:

Thank you very much for your comment. According to your previous comment, we updated the explanation of Fig. 2(b) in the revised manuscript. Both (a) and (b) are included in the Figure 2. The changes are highlighted in the revised manuscript with blue color (See page-15,16, lines-280-290).

---

## [Editor Report · Decision Letter 2]

23 Jan 2023

Women's multidimensional empowerment index and essential newborn care practice in Bangladesh: the mediating role of skilled antenatal care follow-ups

PONE-D-22-01899R2

Dear Dr. Kanchan Kumar Sen, 

We’re pleased to inform you that your manuscript has been judged scientifically suitable for publication and will be formally accepted for publication once it meets all outstanding technical requirements.

Kind regards,

Gouranga Lal Dasvarma, PhD

Academic Editor

PLOS ONE
---

## [Editor Report · Acceptance letter]

27 Jan 2023

PONE-D-22-01899R2 

Women's multidimensional empowerment index and essential newborn care practice in Bangladesh: the mediating role of skilled antenatal care follow-ups 

Dear Dr. Sen:

I'm pleased to inform you that your manuscript has been deemed suitable for publication in PLOS ONE. Congratulations! Your manuscript is now with our production department. 

Kind regards, 

on behalf of

Dr. Gouranga Lal Dasvarma 

Academic Editor

PLOS ONE